# Ultraviolet photolysis of H₂S and its implications for SH radical production in the interstellar medium

Jiami Zhou[1,2,7], Yarui Zhao[1,3,7], Christopher S. Hansen[4,7], Jiayue Yang[1,7], Yao Chang [1], Yong Yu[1], Gongkui Cheng[1], Zhichao Chen[1], Zhigang He[1], Shengrui Yu[2], Hongbin Ding[3], Weiqing Zhang[1], Guorong Wu [1], Dongxu Dai[1], Colin M. Western [5], Michael N.R. Ashfold [5✉], Kaijun Yuan [1✉] & Xueming Yang [1,6✉]

Hydrogen sulfide radicals in the ground state, SH(X), and hydrogen disulfide molecules, H₂S, are both detected in the interstellar medium, but the returned SH(X)/H₂S abundance ratios imply a depletion of the former relative to that predicted by current models (which assume that photon absorption by H₂S at energies below the ionization limit results in H + SH photoproducts). Here we report that translational spectroscopy measurements of the H atoms and S($^1$D) atoms formed by photolysis of jet-cooled H₂S molecules at many wavelengths in the range $122 \leq \lambda \leq 155$ nm offer a rationale for this apparent depletion; the quantum yield for forming SH(X) products, $\Gamma$, decreases from unity (at the longest excitation wavelengths) to zero at short wavelengths. Convoluting the wavelength dependences of $\Gamma$, the H₂S parent absorption and the interstellar radiation field implies that only ~26% of photoexcitation events result in SH(X) products. The findings suggest a need to revise the relevant astrochemical models.

[1] State Key Laboratory of Molecular Reaction Dynamics, Dalian Institute of Chemical Physics, Chinese Academy of Sciences, 457 Zhongshan Road, 116023 Dalian, China. [2] Hangzhou Institute of Advanced Studies, Zhejiang Normal University, 311231 Hangzhou, China. [3] School of Physics, Key Laboratory of Materials Modification by Laser, Ion and Electron Beams, Chinese Ministry of Education, Dalian University of Technology, 116024 Dalian, China. [4] School of Chemistry, University of New South Wales, Sydney, NSW 2052, Australia. [5] School of Chemistry, University of Bristol, Bristol BS8 1TS, UK. [6] Department of Chemistry, Southern University of Science and Technology, 518055 Shenzhen, China. [7] These authors contributed equally: Jiami Zhou, Yarui Zhao, Christopher S. Hansen, Jiayue Yang. ✉email: mike.ashfold@bristol.ac.uk; kjyuan@dicp.ac.cn; xmyang@dicp.ac.cn

Sulfur is one of the more abundant elements in the Universe (S/H $\sim 1.3 \times 10^{-5}$ in the solar photosphere[1], but the abundances of known sulfur-containing molecules in the interstellar medium (ISM) sum to well below this value[2]. Estimates based on the limited number of S-containing compounds detected in diffuse clouds imply a sulfur abundance in such low-density regions not far from the cosmic value[3]. However, the abundance of S-containing species in the external layers of the photodissociation region (PDR) in the Horsehead nebula, for example, is deduced to be ~4-fold depleted (cf. the cosmic value)[4] and one or more orders of magnitude depletions have been reported in cold molecular clouds[5]. Such extreme depletions have been reported not only in cold pre-stellar cores (where most of the sulfur may be locked in icy grain mantles)[6], but also in hot cores/corinos[7,8] where one might expect much of the sulfur to have returned to the gas phase in hot cores and strong shocks—though we note that such conclusions can be very sensitive to assumptions regarding the chemical composition prior to collapse[9]. Given the high-hydrogen abundances and the mobility of hydrogen in the ice matrix, sulfur atoms impinging on interstellar ice mantles are expected to preferentially form $H_2S$, the chemical- and photo-desorption of which constitutes a source of gas phase $H_2S$[10,11].

Commonly used astrochemical models[12,13] assume that photon absorption by $H_2S$ results in dissociation to H + SH products or, at energies above the first ionization potential ($84,432 \pm 2$ cm$^{-1}$)[14], to ionization with relative probabilities determined by the respective photoabsorption and photoionization cross-sections[15]. Interstellar SH radicals were first detected in the atmosphere of the S-type star R Andromedae by infrared (IR) spectroscopy[16], and subsequently identified by their electronic absorption in the solar atmosphere[17] and in translucent interstellar clouds[18], but not detected by rotational spectroscopy until 2012, via the lowest energy $^2\Pi_{3/2}$; $J = 5/2 \leftarrow 3/2$ transition at 1.383 THz[19]. Early analyses returned SH/$H_2S$ abundance ratios of ~13% that were very much smaller than predicted by standard models for SH and $H_2S$ production in turbulent dissipation regions and shocks[19]. Much of the apparent mismatch was subsequently traced to a computational error, but the sense of the discrepancy was deduced still to be valid [3].

The photodissociation of $H_2S$ has been the subject of many prior experimental studies. $H_2S$ displays a weak continuous absorption at wavelengths $\lambda < 260$ nm and numerous stronger absorption features at $\lambda < 155$ nm associated with excitations to Rydberg states[20–22]. Prior photodissociation studies in the long wavelength continuum[23] and at $\lambda = 157.6$ nm[24] revealed prompt S–H bond fission and formation of ground ($X^2\Pi$) state SH radicals, whereas the SH radicals formed at the Lyman-$\alpha$ photolysis wavelength ($\lambda = 121.6$ nm) are exclusively in the excited $A^2\Sigma^+$ state[20,25]. Equations (1)–(5) show the thermochemical thresholds for various spin-allowed fragmentation processes of interest, derived using literature values for the bond dissociation energies $D_0°$(HS–H)[26], $D_0°$(S–H)[27], $D_0°$(H–H)[28] and the electronic term values $T_{00}$(SH(A–X)[29], and $\Delta E$(S($^1D_2$–$^3P_2$))[30] detailed in Supplementary Table 1.

$$H_2S \rightarrow H + SH(X^2\Pi_{3/2}, v = 0, N = 1) \quad \Delta E = 31,451 \pm 4 \text{cm}^{-1} \tag{1}$$

$$\rightarrow H + SH(A^2\Sigma, v' = 0, N' = 0) \quad \Delta E = 62,284 \pm 4 \text{cm}^{-1} \tag{2}$$

$$\rightarrow H + H + S(^3P_2) \quad \Delta E = 60,696 \pm 25 \text{cm}^{-1} \tag{3}$$

$$\rightarrow H + H + S(^1D_2) \quad \Delta E = 69,935 \pm 25 \text{cm}^{-1} \tag{4}$$

$$\rightarrow H_2(X^1\Sigma_g^+, v'' = 0, J'' = 0) + S(^1D_2) \quad \Delta E = 33,817 \pm 25 \text{cm}^{-1} \tag{5}$$

In addition, there was an earlier report of isotopic fractionation in solid sulfur films deposited following broadband (~2 nm), non-collision-free photolysis of slowly flowing samples of $H_2S$ gas at several VUV wavelengths[31].

Herein, we present a photochemical rationale for the deduced paucity of SH radicals in PDRs of the ISM based on the predissociation of most of the primary SH photoproducts. The study employed the intense, pulsed vacuum ultraviolet (VUV)-free electron laser (FEL) at the Dalian coherent light source (DCLS) to measure product state-resolved translational energy spectra of both the H and S($^1D$) atom photoproducts formed following tuneable VUV photolysis of $H_2S$ using, respectively, the H-atom Rydberg tagging and velocity map ion imaging techniques. The experimental results imply that only ~26% of photoexcitation events result in SH(X) products.

## Results and discussion

**Product translational energy distributions.** H atom time-of-flight (TOF) spectra were recorded following photolysis of a jet-cooled $H_2S$ sample at many wavelengths resonant with its discrete VUV absorption peaks. These spectra were converted to the corresponding H atom translational energy distributions, and momentum conservation arguments were then used to derive the total translational energy distributions, $P(E_T)$, where

$$E_T = \frac{1}{2} m_H \left(\frac{d}{t}\right)^2 (1 + m_H/m_{SH}), \tag{6}$$

$m$ is the photofragment mass, $d$ is the distance separating the interaction region and the detector, and $t$ is the H atom TOF measured over this distance. Figure 1 shows $P(E_T)$ spectra obtained at $\lambda$ = (a) 154.53 nm (64712 cm$^{-1}$), (b) 139.11 nm (71886 cm$^{-1}$), and (c) 122.95 nm (81334 cm$^{-1}$). The polarization vector of the photolysis laser radiation ($\varepsilon_{phot}$) used to record these spectra was aligned at the magic angle ($\theta = 54.7°$) to the detection axis, thereby ensuring that the spectra are insensitive to the product channel dependent recoil anisotropies revealed by spectra taken with $\varepsilon_{phot}$ aligned, respectively, parallel ($\theta = 0°$) and perpendicular ($\theta = 90°$) to the detection axis. The inset in Fig. 1b illustrates such recoil anisotropy for the products of photolysis at $\lambda = 139.11$ nm. H atom TOF spectra recorded at all three wavelengths with $\varepsilon_{phot}$ aligned at $\theta = 0°$ and 90° are shown in Supplementary Fig. 1, but detailed descriptions of the rovibronic level structure of the SH radical and of the photofragmentation dynamics responsible for the populations and recoil anisotropies of the various SH products formed when exciting $H_2S$ at many different wavelengths in the range $122 \leq \lambda \leq 155$ nm are reserved for a future publication. Here we choose to highlight just the following features:

i. The SH fragments formed at $\lambda = 154.53$ nm are in their ground (X) state, reinforcing the findings of prior photofragmentation studies at $\lambda = 157.6$ nm[24] and at all longer wavelengths[23]. The combs festooned above the spectrum show that the SH(X) population is distributed over a wide range of quantum states, spanning rotational (N) levels of most if not all vibrational (v) levels. The SH(X) potential energy curve correlates with H + S($^3P$) fragments at infinite separation and the small feature at low $E_T$ in Fig. 1a is consistent with three-body dissociation to 2 H + S ($^3P$) products, the threshold energy for which ($\Delta E$(3)) lies ~1600 cm$^{-1}$ below that for two-body dissociation to H + SH(A) products.

ii. $\lambda = 139.11$ nm photolysis yields both SH(X) and SH(A) fragments. The latter are revealed by the sharp step at $E_T = E_{phot} - \Delta E(2)$, where $E_{phot}$ is the photolysis photon energy and $\Delta E(2)$ is the threshold energy for forming H + SH(A)

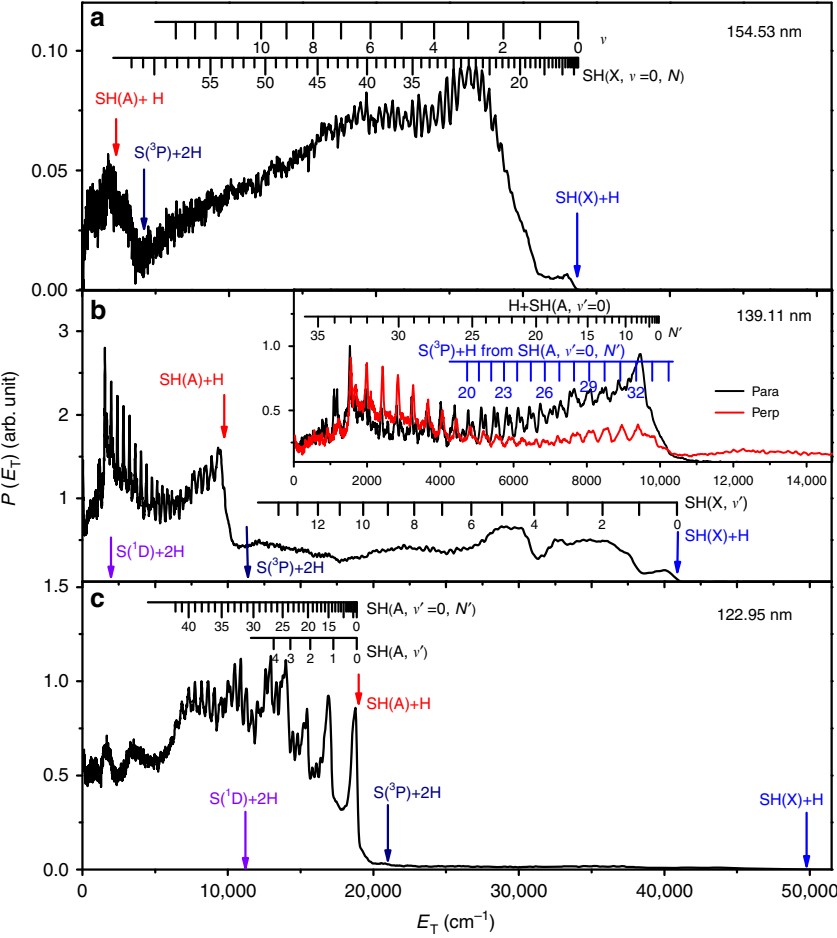

**Fig. 1 Translational energy spectra from H₂S photodissociation.** $P(E_T)$ spectra derived from H atom time-of-flight spectra following photodissociation of H₂S at **a** 154.53 nm, **b** 139.11 nm, and **c** 122.95 nm with the detection axis aligned at the magic angle ($\theta = 54.7°$) to the polarization vector of the photolysis laser radiation ($\varepsilon_{phot}$). The inset in **b** shows an expanded view of the low $E_T$ part of the corresponding $\theta = 0°$ and 90° data. The combs on these spectra show the $E_T$ values associated with formation of H atoms in conjunction with selected rovibrational levels of the primary SH(X) and SH(A) fragments and, in **b**, with H atoms formed by predissociation of primary (SH(A, $v' = 0$, $N'$) fragments. The maximum $E_T$ values associated with each of channels (1)-(4) are shown by vertical blue, red, navy, and violet arrows, respectively; the navy arrow indicates the $E_{phot} - \Delta E(3)$ value at which we separate the spectrum into primary H + SH(X) and (direct or indirect) three-body product formation. Source data are provided as a Source Data file.

products. The SH(A) fragments formed at this wavelength are mainly in their $v' = 0$ level but carry high levels of rotational excitation. $\lambda = 149.15$ nm was the longest photolysis wavelength at which structure unambiguously attributable to H + SH(A) products was observed.

iii. SH(A) fragments dominate when exciting at $\lambda = 122.95$ nm, as at 121.6 nm[25]. The combs in Fig. 1c show that the SH(A) fragments are formed mostly in low $v$ levels but have rotational state distributions that extend to the highest $N'$ levels supported by the SH(A) potential energy function[25]. Note that these highest $N'$ levels have rotational energies well in excess of the bond dissociation energy of the SH(A) state—indicated by the S(¹D) + 2 H arrow in Fig. 1c—and that the signal at lowest $E_T$ in Fig. 1c must include contributions from direct three-body-dissociation to such products. Population of similar "super-rotor" levels (i.e., rotational levels that are only quasi-bound by virtue of the centrifugal barrier associated with the high-rotational angular momentum) has also been reported recently in the case of OH(X) products arising in the VUV photolysis of H₂O[32].

iv. SH(A) state radicals predissociate on a nanosecond (or shorter) timescale to yield H + S(³P$_J$) atom products[33]. Thus, every primary SH(A) photofragment will dissociate

to yield a second H atom during the time that the Rydberg tagging laser radiation is present. Such secondary H atoms must contribute to the measured H atom TOF spectra and the $P(E_T)$ spectra derived therefrom. The predissociation of SH(A) radicals favors population of the ground ($J = 2$) spin-orbit state of the S(³P$_J$) product[33], and a comb indicating the expected $E_T$ values of H + S(³P₂) products from predissociation of different SH(A, $v' = 0$, $N'$) levels formed in the $\lambda = 139.11$ nm photolysis of H₂S is also included in the inset to Fig. 1b.

The term ($E_{phot} - \Delta E(3)$), henceforth written as $E_T(3)$, will prove to be a useful point at which to divide $P(E_T)$ spectra such as those shown in Fig. 1. All signal measured at $E_T > E_T(3)$ is associated with branching into H + SH(X) products, while signal at $E_T < E_T(3)$ must be associated with three-body dissociation, yielding two H atoms—either directly, or indirectly when the second H atom comes from the predissociation of a primary SH(A) photoproduct. Thus, we can define the following fraction, $\Gamma$, for forming SH(X) products following VUV photoexcitation of H₂S

$$\Gamma = \frac{\sum {}^P(E_T > E_T(3))}{\sum {}^P(E_T > E_T(3)) + 0.5 \sum {}^P(E_T < E_T(3))} \quad (7)$$

where the 0.5 in the second term in the denominator recognizes that, for this fraction of the product yield, the absorption of one photon yields two detectable H atom products. This partitioning would not be perfect if, for example, dissociation at some wavelengths resulted in a (small) yield of SH(X) super-rotor states, but it allows a good estimation of the wavelength dependence of $\Gamma$. The implications of the derived $\Gamma(\lambda)$ function for interstellar SH number densities are considered later.

**Imaging the $S(^1D) + H_2$ products.** Figure 2a shows a time-sliced velocity map image of the $S(^1D)$ atoms formed following photolysis of a jet-cooled $H_2S$ sample at $\lambda = 139.11$ nm, with $\varepsilon_{phot}$ aligned in the plane of the image (as indicated by the red arrow) and subsequent resonant ionization by absorption of a single $\lambda = 130.091$ nm photon. The latter photons were generated by four wave difference frequency mixing, by overlapping the focussed outputs of two table-top dye lasers operating at $\lambda = 212.556$ nm and $\lambda = 580.654$ nm in a Kr/Ar gas mixture. Image analysis yields the velocity distribution, and thence the corresponding $P(E_T)$ distribution shown in Fig. 2b assuming momentum conservation and that the partner fragment is $H_2$. The image appears isotropic. The $P(E_T)$ spectrum is structured and peaks at $E_T \sim 0$, demonstrating preferential population of high $v''$ and $J''$ levels of the $H_2$ fragments (henceforth represented as $H_2^\#$)—as shown by the combs above the spectrum in Fig. 2b. This finding confirms and extends conclusions reached in an early resonance enhanced multiphoton ionization study of the $H_2$ molecules formed via two-photon dissociation of $H_2S$ at similar overall excitation energies[34].

The $H_2(X)$ potential correlates with two ground state H atoms at infinite separation and, recalling Eq. (4), the energetic threshold for the triple fragmentation of $H_2S$ to $2H + S(^1D)$ products is ~1960 cm$^{-1}$ below the energy provided by a 139.11 nm photon. Any such three-body dissociation should thus be revealed by signal appearing with $E_T < 1960$ cm$^{-1}$ in Fig. 2. This spectrum shows no obvious step or discontinuity at low $E_T$ values, but such signal is apparent (as a bright central spot) in $S(^1D)$ images recorded at shorter excitation wavelengths. For now, we estimate the possible quantum yield of the $2H + S(^1D)$ three-body dissociation channel as follows: The ratio of the signal appearing with $E_T$ values that are, respectively, below and above $(E_{phot} - \Delta E(4))$ in spectra such as that shown in Fig. 2b provides a relative measure of the branching into channels (4) and (5), i.e., into $2H + S(^1D)$ versus $H_2 + S(^1D)$ products. Such a partitioning

in the case of Fig. 2b returns a ratio of ~0.25, which must be viewed as an upper limit, since it makes no allowance for any primary fragmentations that yield $H_2$ products in super-rotor levels. The slow H atom products from channel (4) must also contribute to the $P(E_T)$ spectrum derived from H atom TOF measurements (Fig. 1b and Supplementary Fig. 1b). Very slow H atoms will be under-sampled in the TOF experiment but, recognizing that channel (4) yields two H atoms, our best analyses suggest that this channel accounts for no >3% of the dissociations yielding H atoms at this wavelength—implying a quantum yield for forming $H_2 + S(^1D)$ products at $\lambda = 139.11$ nm $\leq 0.12$ and that S–H bond fission is the dominant primary event at this wavelength.

**The fragmentation dynamics.** As noted previously, detailed discussion of the fragmentation dynamics revealed by the present measurements will be reserved for a future publication, but many aspects of the observed product energy disposal are seen to be qualitatively similar to those found in previous studies of the lighter analog, $H_2O$[35]. As in $H_2O$, the two key valence absorption continua that support HS–H bond fission arise from excitation of an electron from the highest occupied molecular orbitals (the HOMO, $3p_x$ and HOMO–1, $3p_z$ (with $b_1$ and $a_1$ symmetry, respectively, in $C_{2v}$)) to the 4s ($a_1$) Rydberg orbital. The 4s orbital acquires progressively greater anti-bonding valence $\sigma^*$ character on stretching one S–H bond. The $^1B_1$ and $^1A_1$ states formed by these electron promotions, which we henceforth label $\widetilde{A}$ and $\widetilde{B}$ to emphasize the parallels with $H_2O$, are a Renner–Teller pair and degenerate at linear geometries. Both correlate with ground state $H + SH(X)$ products at linear geometry (and the ground $\widetilde{B}^1A_1$ state of $H_2S$ correlates with excited $H + SH(A)$ products). However, the $\widetilde{A}$ and $\widetilde{B}$ states split apart on bending. The $\widetilde{A}$ state potential energy surface (PES) is relatively flat with respect to changing the interbond angle, $\angle$HSH, but the $\widetilde{B}$ state potential rises in energy and its crossings with the $\widetilde{X}$ state PES at extended HS–H (and H–HS) bond lengths develop into conical intersections (CIs) upon bending away from linearity. At all non-linear geometries, therefore, the $\widetilde{B}$ state PES correlates with electronically excited SH(A) products.

The ground state and the Rydberg (R) states of current interest all have bent equilibrium geometries ($\angle$HSH $\sim 92°$). The R states are diabatically bound with respect to HS–H bond extension but,

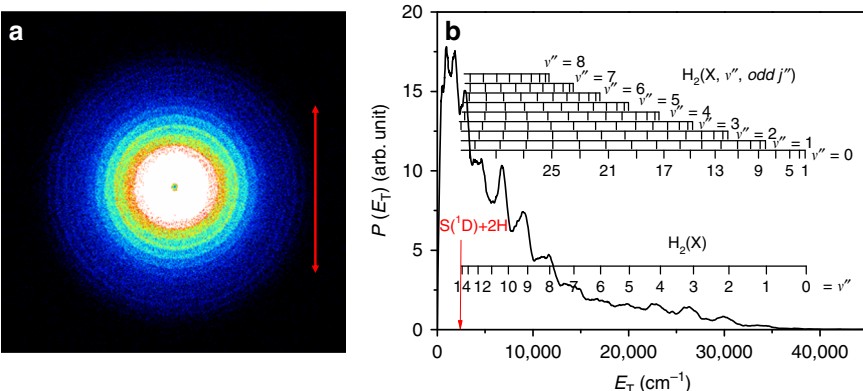

**Fig. 2 Velocity map imaging and translational energy spectrum from H₂S photodissociation. a** Time-sliced velocity map image of the S($^1D_2$) photofragments from photolysis of $H_2S$ at $\lambda = 139.11$ nm with the polarization vector of the photolysis laser radiation ($\varepsilon_{phot}$, shown by the doubled headed red arrow) aligned in the plane of the image. **b** $P(E_T)$ spectrum derived therefrom assuming recoil from an $H_2$ co-fragment. The superposed combs indicate the $H_2(v'', J'')$ states responsible for the evident structure with, for simplicity, only the *ortho-* (odd) $J''$ states indicated. The red arrow indicates the maximum $E_T$ values associated with the S($^1D$)+2H channel. Source data are provided as a Source Data file.

as with $H_2O$ molecules excited to the $\widetilde{C}$ $^1B_1$ state[36-38], $H_2S$ molecules excited to such Rydberg states can predissociate by coupling to the $\widetilde{A}$ and/or $\widetilde{B}$ state continua. Given the relative energetics and the topographies of the respective PESs[39-41], $H_2S$ (R) molecules prepared at longer excitation wavelengths would be expected to dissociate via the former, lower energy pathway to yield H + SH(X) products in a range of $\nu$ levels but with relatively modest rotational excitation—in accord with experimental observation at $\lambda = 154.53$ nm, for example.

$H_2S(R)$ molecules prepared by excitation at shorter wavelengths can couple to the $\widetilde{B}$ state PES also, which supports much richer fragmentation dynamics. The $\widetilde{B}$ state PES (illustrated as a contour plot in Fig. 3 as functions of $\angle HSH$ and one S–H bond length, $R_{S-H}$, with the other held fixed at its ground state equilibrium value of 2.5 bohr) encourages both HS–H bond extension and opening of $\angle HSH$ towards linearity. Molecules that reach linear geometries at HS–H bond lengths less than or equal to the $R_{S-H}$ value of the $\widetilde{B}/\widetilde{X}$ CI that we henceforth term CI1 (labeled type I trajectories in Fig. 3) can transition to the $\widetilde{A}$ state PES (by $\widetilde{B}/\widetilde{A}$ Renner–Teller coupling) or the $\widetilde{X}$ state PES (by nonadiabatic coupling at CI1) and dissociate to H atoms, together with highly rotationally excited SH(X) products. Other $H_2S(R)$ molecules that couple to the $\widetilde{B}$ state PES will fail to attain linearity before reaching this critical $R_{S-H}$ distance, however. These molecules remain on the $\widetilde{B}$ state PES and can dissociate to H +

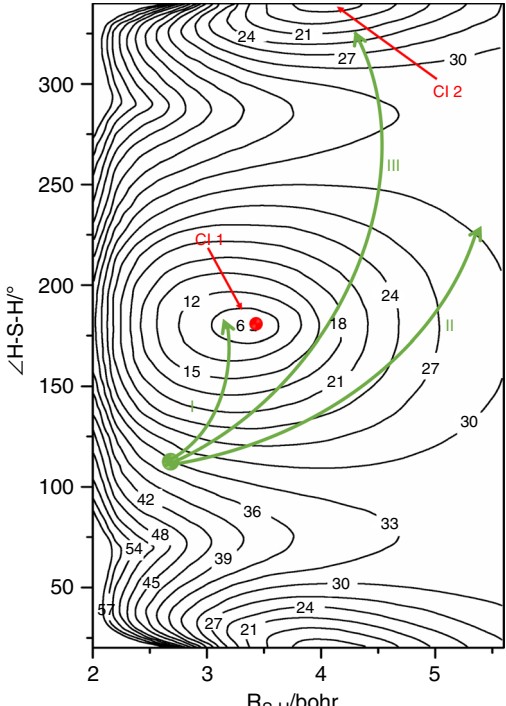

**Fig. 3 Potential energy surface contour plots of $H_2S$.** Potential energy surface of the $\widetilde{B}$ state of $H_2S$ plotted as a function of $R_{S-H}$ (the distance of the departing H atom from the S atom, with the other S–H bond held fixed at the ground state equilibrium bond length (2.5 bohr)) and the interbond angle $\angle HSH$. The contour labels are given in units of $10^3$ cm$^{-1}$, with the energy zero calculated at $R_{S-H} = 100$ bohr (i.e., the H + SH(X) dissociation limit). The two regions of conical intersection with the $\widetilde{X}$ state PES at linear H-S-H and S-H-H geometries (CI1 and CI2, respectively) are highlighted in red. The green curves show illustrative type I, II and III trajectories originating from the region sampled by vertical Franck–Condon excitation at the ground state equilibrium geometry. Source data are provided as a Source Data file.

(highly rotationally excited) SH(A) products via type II trajectories—as observed following excitation at $\lambda = 139.11$ and 122.95 nm. In other cases, the separating H atom may "slingshot" around the heavy S atom and explore the second $\widetilde{B}/\widetilde{X}$ CI at linear S–H–H geometries (henceforth termed CI2) having followed a type III trajectory. Analogy with $H_2O$[42-44] suggests that such type III trajectories are likely to offer another route to highly vibrationally and/or rotationally excited SH(X) fragments and to $S(^1D)$ + $H_2$ products.

These various fragmentation pathways are all illustrated schematically in Fig. 3 for the case of $H_2S$ excitation with a VUV photon of intermediate energy, e.g., 139.11 nm. The topography of the $\widetilde{B}$ state PES encourages high-rotational excitation in the SH(A) products and rovibrational excitation in the $H_2^{\#}$ products from the respective two-body fragmentations—outcomes that are accentuated further by tuning to shorter wavelengths. The present discussion suggests that three-body dissociation to $2H + S(^1D)$ products at shorter photolysis wavelengths can arise via two distinct mechanisms involving, respectively, initial distortions towards H----SH(A) products in the vicinity of CI1 and towards S----$H_2^{\#}$ products around CI2.

**Implications for SH densities in the interstellar medium.** The present imaging studies show that the $S(^1D)$ + $H_2$ product channel is active at $\lambda = 139.11$ nm (and at neighboring wavelengths), but with a rather small quantum yield, thus confirming that S–H bond fission is the dominant primary event following photoexcitation of $H_2S$ at VUV wavelengths. However, S–H bond fission can result in several final outcomes (channels (1)–(4)), only one of which yields stable SH(X) products. The H-atom Rydberg tagging studies show $\Gamma$ decreasing from 1 (at $\lambda \geq 157.6$ nm) to 0 (by $\lambda \leq 121.6$ nm). Figure 4 collects together the wavelength dependences of: $\Gamma$ determined in the present study; the general interstellar radiation field (ISRF, from Draine[45] and extended according to van Dishoeck and Black[46]); and the total photoabsorption[20] and photoionization cross-sections[15] of the parent $H_2S$ molecule ($\sigma_{tot}$ and $\sigma_{ion}$, respectively) down to the wavelength corresponding to the ionization threshold of atomic H ($\lambda = 91.2$ nm)[30]. Note, the very short lived nature of the excited states populated at all wavelengths implies that the photodissociation and photoabsorption cross-sections are identical at all $\lambda > 118.4$ nm.

The peak of the ISRF matches with the region of strong parent absorption, within which $\Gamma$ is changing rapidly. Reference to Fig. 4 shows that absorption at wavelengths below the ionization limit ($\lambda \leq 118.4$ nm) will increasingly result in ionization, but any dissociative ionization at $\lambda > 91.2$ nm will yield $S^+$ or $SH^+$ products (i.e., no neutral SH(X) products)[15]. It is thus reasonable to assume $\Gamma = 0$ at all wavelengths in Fig. 4 shorter than those studied in the present work, and to estimate the fraction $\phi$ of all photoexcitation events that result in SH(X) fragments by convoluting $\Gamma$ with the standard model of the ISRF and with $\sigma_{tot}$. The uncertainty in $\Gamma$ can be quantified (12.5%, defined as the normalized (by range) root-mean-square error from the fitted sigmoidal function), but to estimate an uncertainty on $\phi$ also requires knowledge of the $\lambda$-dependent uncertainties associated with the ISRF and with $\sigma_{tot}$. These are not available, but are likely dominated by the uncertainty in the form of the ISRF, which is quoted as being "within about 50%"[47]. From all the foregoing, $\phi$ is definitely «1, with a best estimate of $\phi$ ~26% based on the wavelength dependent functions displayed in Fig. 4. The largest source of uncertainty in $\phi$ is likely to be that associated with the ISRF, which is necessarily high for it to represent the galactic average. However, the remaining quantities are invariant for $H_2S$ molecules anywhere in the universe and a more precise value of $\phi$

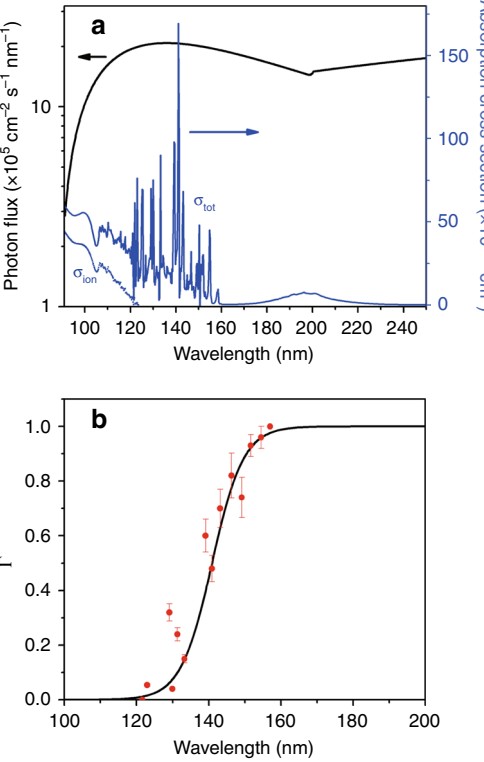

**Fig. 4 The wavelength dependences of SH(X) quantum yields.** Plot showing the wavelength dependences of **a** the general interstellar radiation field (ISRF, the black line) and the total absorption ($\sigma_{tot}$, the blue line) and photoionization ($\sigma_{ion}$, the blue dotted line) cross-sections of $H_2S$ and **b** the quantum yield for forming SH(X) photoproducts, $\Gamma$ (Eq. (7)), determined in the present work (the red dots). The sigmoidal function (the black line) through the latter data is used to derive the reported overall SH(X) product quantum yield. The error bars represent the standard deviation of three independent measurements. Source data are provided as a Source Data file.

can be determined for comparison with a particular observation by employing a better representation of the radiation field specific to the region of space under investigation. The Supplementary Information includes a directory containing a Python script and the necessary data files to determine $\phi$ for any supplied ISRF. Note, any such analysis is still likely to return an upper limit on the branching into SH(X) fragments, as it makes no allowance for the (small but non-zero) branching into $H_2$ + S($^1$D) products.

The SH(X) radicals detected in the ISM are in low $N$ levels of the ground ($v = 0$) state. Such products dominate when $H_2S$ is photolyzed in the weak long wavelength continuum (e.g., at $\lambda > 193$ nm)[23], but the SH(X) fragments formed at shorter photolysis wavelengths span a wide range of higher $v$, $N$ levels (as shown in Fig. 1a, b. Given the low pressures (and collision frequencies) prevailing in the ISM, such internally excited photoproducts will relax radiatively, with rates that can be found in the recently published ExoMol list of SH rotational, rovibrational and rovibronic term values and transitions[48]. Supplementary Fig. 2, which shows the calculated energy level dependent radiative lifetimes, implies ~0.1 s lifetimes for the range of SH(X, $v$, $N$) levels populated by shorter wavelength photolysis of $H_2S$. The decay of such population to the $v = 0$, low $N$ levels identified in the ISM will involve radiative cascade via tens (or more) of transitions but, in total, should be complete within minutes (Supplementary Note 1). Measurements of interstellar SH radicals using the GREAT instrument on SOFIA[19] led to initial SH/$H_2S$

abundance ratio estimates of ~13%—much smaller than that predicted by standard astrochemical models. The present study clearly identifies competition from hitherto under-appreciated three-body dissociation as an inevitable source of SH(X) radical depletion (cf. $H_2S$) in the ISM. The findings may need to be added into the related astrochemical models.

## Methods

**Vacuum ultraviolet free electron laser (VUV-FEL) radiation.** The experiment employs a recently constructed apparatus for molecular photochemistry centered on a VUV-FEL facility[49] operating in the high gain harmonic generation (HGHG) mode[47], wherein a seed laser is injected to interact with the electron beam in the modulator. The seed laser pulse (in the range $240 < \lambda_{seed} < 360$ nm) is generated from a picosecond (ps) Ti:sapphire laser. The electron beam is generated from a photocathode radio frequency gun and accelerated to a beam energy of ~300 MeV by 7 S-band accelerator structures, with a bunch charge of 500 pC. The micro-bunched beam is sent through the radiator which is tuned to the $n$th harmonic of the seed wavelength and coherent FEL radiation with wavelength $\lambda_{seed}/n$ is emitted. Optimizing the linear accelerator results in a high-quality beam with emittance ~1.5 mm·mrad, a projected energy spread of ~1‰, and pulse duration of ~1.5 ps. The VUV-FEL currently operates at 10 Hz, the maximum pulse energy is ~500 µJ/pulse and with a specified continuous tunability range of $50 < \lambda < 150$ nm and a typical spectral bandwidth of ~50 cm$^{-1}$. However, as the present study shows, the stated long wavelength limit can be slightly exceeded.

The H-atom Rydberg tagging TOF technique, pioneered by Welge and coworkers[50], employs a sequential two-step excitation of the H atom on, first, the $n = 2 \leftarrow n = 1$ transition by absorption of a 121.6 nm photon, then by UV laser excitation (365 nm) from the $n = 2$ level to a high-$n$ Rydberg state. The neutral Rydberg atoms fly a distance $d \sim 280$ mm to a rotatable microchannel plate (MCP) detector where they are field ionized, giving a TOF temporal resolution of <0.5%. The sample beam was generated by expanding a mixture of 1% $H_2S$ and Ar at a stagnation pressure of 600 Torr through a 0.5 mm-diameter pulsed nozzle. The molecular beam crosses the VUV-FEL beam at right angles. The polarization of the VUV-FEL pulse is fixed in the horizontal plane, so "parallel" and "perpendicular" TOF spectra are obtained by rotating the MCP detector about the VUV-FEL propagation axis. The 121.57 nm tagging laser beam generates a background H atom spectrum also, which is measured by recording spectra with the VUV-FEL beam on and off and subtracted accordingly.

The apparatus and most of the procedures for the velocity map imaging (VMI) experiments have been described in the Supplementary Fig. 3 and Supplementary note 2[51,52]. The pulsed supersonic beam was again generated by expanding a mixture of 1% $H_2S$ and Ar into the source chamber where it was skimmed before entering (through a 2 mm hole in the first electrode), and propagating along the center axis of, the ion optics assembly mounted in the same differentially pumped reaction chamber. The molecular beam was intersected at right angles by the photolysis and probe laser beams between the second and the third plates of the ion optics assembly. The photolysis photons were provided by the FEL with, again, $\varepsilon_{phot}$ fixed in the horizontal plane and thus parallel to the front face of the detector. The S($^1$D$_2$) photoproducts are probed by one photon excitation at $\lambda = 130.091$ nm, which populates the autoionizing $3p^3(^2D^o)5s$; $^1D_2{}^o$ level. These latter photons were generated by four wave difference frequency mixing the frequency doubled output from one dye laser (at $\lambda = 212.556$ nm) with the fundamental output of a second dye laser (at $\lambda = 580.654$ nm) in a Kr/Ar gas mixture. The S$^+$ ($^1D_2{}^o$) ions are accelerated through the remaining ion optics and a 740 mm long field-free region before impacting on a 70 mm-diameter chevron double MCP detector coupled with a P43 phosphor screen. Transient images on the phosphor screen were recorded by a charge-coupled device (CCD) camera, using a 30 ns gate pulse voltage in order to acquire time-sliced images. Further details of the FEL-VMI experiment are provided in the Supplementary Information.

## Data availability

The data supporting this study are available from the authors on reasonable request. The source data underlying Figs. 1–4 are also provided as a Source Data file.

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

## Acknowledgements

The experimental work is supported by the Strategic Priority Research Program of the Chinese Academy of Sciences (Grant No. XDB17000000), the Chemical Dynamics Research Center (Grant No. 21688102), the National Natural Science Foundation of China (NSFC Nos. 21873099, 21673232, 21673234, 21922306), and the international partnership program of Chinese Academy of Sciences (No. 121421KYSB20170012). C.S.H. acknowledges receipt of an Australian Research Council Discovery Early Career Award (DE200100549). M.N. R.A. gratefully acknowledges funding from the Engineering and Physical Sciences Research Council (EPSRC, EP/L005913) and both C.S.H. and M.N.R.A. are grateful to the NSFC Center for Chemical Dynamics for the award of Visiting Fellowships. The authors thank the Dalian Coherent Light Source for experimental technique supports.

## Author contributions

K.J.Y., M.N.R.A., and X.M.Y. designed the experiments. J.M.Z., Y.R.Z., C.S.H., Y.C., J.Y.Y., Y.Y., G.K.C., Z.C.C., and Z.G.H. performed the experiments. K.J.Y., M.N.R.A., C.M.W., J.M.Z., Y.R.Z., and C.S.H. analyzed the data and C.S.H. prepared the Python scripts included in the Supplementary Information. K.J.Y., M.N.R.A., C.S.H., Z.C.C., S.R.Y., H.B.D., W.Q.Z., G.R.W., D.X.D., and X.M.Y. discussed the experimental results. K.J.Y., M.N.R.A., C.S.H., and X.M.Y. prepared the manuscript.

## Competing interests

The authors declare no competing interests.
