## [Peer Review File · Nature Communications]

Reviewers' comments:

Reviewer #1 (Remarks to the Author):

Please find below my referee's report for the manuscript "Ultraviolet Photolysis of H₂S: Implications for SH Radical Density Estimate in the Interstellar Medium" by J. Zhou et al.. I recommend the paper for publication in Nature Communications as it stands.

The new results – concerning the yield of ground-state SH radicals following VUV photolysis of hydrogen sulfide – are highly relevant to photochemical models of sulfur-bearing molecules in the interstellar medium. The new experimental results are used to show that the ground state SH yield is strongly wavelength dependent and, in fact, effectively goes to zero at wavelengths shortward of about 125 nm. When coupled with a standard model of the radiation field in the ISM and H₂S photoabsorption cross sections, the wavelength-dependent photolysis yield nicely accounts for the observed depletion of SH relative to H₂S in astronomical sources. In addition to the astrophysical applications that rely on a quantitative understanding of the H₂S dissociation paths following VUV excitation, there is intrinsic merit on the basic molecular physics side of the ledger in elucidating these dissociation mechanisms. The paper is extremely well written and comprehensive; the experimental methods are presented at an appropriate level of detail should other researchers consider trying to reproduce the results. The experimental techniques, which utilize a pulsed VUV free-electron laser, velocity map imaging, and H-atom tagging, are state of the art. In short, this is an excellent piece of experimental work that is well-presented and whose results are applied to at least partly resolve an existing discrepancy between astronomical observations and photochemical models.

I have only one question for the authors: In figure 4, the total photoabsorption cross section of H₂S is plotted as a function of wavelength. That cross section is used in the convolution calculation that results in the yield of ground state SH. I believe that it is assumed in this calculation that the H₂S photodissociation cross section is identical to the H₂S photoabsorption cross section in this wavelength region. Is this known to be the case? There appears to be sharp structure in the photoabsorption cross section (at least at the level of resolution in the figure); that might indicate a non-zero probability of radiative relaxation of H₂S, which would also lower the yield of SH. It would be helpful if the authors could clarify this one point.

Reviewer #2 (Remarks to the Author):

I was asked to provide a report on the paper entitled "Ultraviolet Photolysis of H₂S: Implications for SH radical density estimates in the interstellar medium". This paper presents experimental results on the photolysis of H₂S by UV photons. The goal here is to constrain the products of the photodissociation of gas-phase H₂S to see if and how much HS is produced in order to explain the observed HS/H₂S abundance ratio in diffuse interstellar medium. In fact, current astrochemical models assume that H₂S photodissociation produces only one channel leading to HS. Using this branching ratio in chemical models produces a larger HS/H₂S than what is observed.

The present paper shows that the photodissociation of H₂S only leads to something like 26% of HS and so concludes that astrochemical models should be changed according to this result and this may explain the low observed HS/H₂S.

Most of the paper focuses on the description of the results of the experiment. Not being an expert on these experiments, I am not able to understand in details what is explained. There is one paragraph on the implication of their experimental result on the study of HS in the interstellar medium. This paragraph is more a discussion than a demonstration that their results will explain the observations as they do not go beyond the estimation of the branching ratio. These new data has not been included in a chemical model and so its impact is speculative.

Overall, such experimental results are crucial for astrochemical studies. These specific results can indeed be important and have an impact. The paper is however difficult to read for a general astrochemist not coming from this field of research. The very promising title of the paper is not completely reflecting the content of the paper.

Reviewer #3 (Remarks to the Author):

The article suggests to revise the relevant astrochemical models to justify the depletion of SH(X) in the interstellar medium. Based on the photolysis of jet cooled H₂S molecules in the 122 - 155 nm range the authors show that the quantum yield to form SH(X) decreases from 1 (at the longest excitation wavelengths) to zero at short wavelengths. Convoluting the interstellar radiation field taken from the literature with the total absorption cross-sections of H₂S the authors report that only 26% of all photoexcitation events should result in SH(X) fragments. The 26% is a very challenging value. What are the error estimates for this value? There are many theoretical spectroscopic studies that describe the VUV photo-dissociation dynamics of H₂S. This paper shows again another UV photolysis of H₂S study but the obtained results cannot be used to estimate the SH radical density in the ISM as claimed by the author. Before publication the authors should pay attention on a few typos such as those on page 32 line 66 and in figure 4. I suggest also that they should change the title

of this article and review the part concerning the SH density in the ISM. They should also link their study to the Chakraborty's work which was not mentioned and related to the sulfur isotopic fractionation in VUV photodissociation of H₂S at 121.6, 139.1, and 157.6 nm, the same wavelengths used in the present study (PNAS 2013 110 (44) 17650-17655).

Response to reviewers' comments

Reviewer #1

Please find below my referee's report for the manuscript "Ultraviolet Photolysis of H₂S: Implications for SH Radical Density Estimate in the Interstellar Medium" by J. Zhou et al.. I recommend the paper for publication in Nature Communications as it stands.

The new results – concerning the yield of ground-state SH radicals following VUV photolysis of hydrogen sulfide – are highly relevant to photochemical models of sulfur-bearing molecules in the interstellar medium. The new experimental results are used to show that the ground state SH yield is strongly wavelength dependent and, in fact, effectively goes to zero at wavelengths shortward of about 125 nm. When coupled with a standard model of the radiation field in the ISM and H₂S photoabsorption cross sections, the wavelength-dependent photolysis yield nicely accounts for the observed depletion of SH relative to H₂S in astronomical sources. In addition to the astrophysical applications that rely on a quantitative understanding of the H₂S dissociation paths following VUV excitation, there is intrinsic merit on the basic molecular physics side of the ledger in elucidating these dissociation mechanisms. The paper is extremely well written and comprehensive; the experimental methods

are presented at an appropriate level of detail should other researchers consider trying to reproduce the results. The experimental techniques, which utilize a pulsed VUV free-electron laser, velocity map imaging, and H-atom tagging, are state of the art. In short, this is an excellent piece of experimental work that is well-presented and whose results are applied to at least partly resolve an existing discrepancy between astronomical observations and photochemical models.

I have only one question for the authors: In figure 4, the total photoabsorption cross section of H₂S is plotted as a function of wavelength. That cross section is used in the convolution calculation that results in the yield of ground state SH. I believe that it is assumed in this calculation that the H₂S photodissociation cross section is identical to the H₂S photoabsorption cross section in this wavelength region. Is this known to be the case? There appears to be sharp structure in the photoabsorption cross section (at least at the level of resolution in the figure); that might indicate a non-zero probability of radiative relaxation of H₂S, which would also lower the yield of SH. It would be helpful if the authors could clarify this one point.

Author Reply: It is safe to assume that H₂S dissociates with unit (or essentially unit) quantum yield at all UV wavelengths longer than the first ionization potential. This was implicit in the submitted paper and has now been made explicit on p. 9 of the revised submission. Photofragmentation studies in the long wavelength continuum return anisotropic H atom recoil velocity distributions consistent with prompt dissociation on a timescale shorter than the parent rotational period (ref. 26), and the present paper confirms that such behaviour extends to shorter (VUV) wavelengths also (Figures 1 and S1). The most long-lived (*i.e.* least heavily predissociated) excited state of H₂S is the state populated when exciting at $\lambda = 139.1$ nm, which shows resolved rovibronic structure in absorption (ref. 25). The sharpest lines in that absorption band show lifetime broadened linewidths of ~ 1.5 cm⁻¹, implying a \sim ps excited state lifetime with respect to predissociation. Given the absorption cross-section (oscillator strength) and excitation energy of the transition, the radiative lifetime of this excited state will be in the ns range. There are no other plausible rival excited state decay channels so, even for this least predissociated excited state of H₂S, the dissociation quantum yield must be >0.999 (*i.e.* essentially unity). Thus it is reasonable to assume that the photodissociation and photoabsorption cross sections of H₂S are identical at all $\lambda > 118.4$ nm. At yet shorter wavelengths, photoionization competes with photodissociation, as discussed in the paper.

Reviewer #2

I was asked to provide a report on the paper entitled "Ultraviolet Photolysis of H₂S: Implications for SH radical density estimates in the interstellar medium". This paper presents experimental results on the photolysis of H₂S by UV photons. The goal here is to constrain the products of the photodissociation of gas-phase H₂S to see if and how much HS is produced in order to explain the observed HS/H₂S abundance ratio in diffuse interstellar medium. In fact, current astrochemical models assume that H₂S photodissociation produces only one channel leading to HS. Using this branching ratio in chemical models produces a larger HS/H₂S than what is observed. The present paper shows that the photodissociation of H₂S only leads to something like 26% of HS and so concludes that astrochemical models should be changed according to this result and this may explain the low observed HS/H₂S.

Most of the paper focuses on the description of the results of the experiment. Not being an expert on these experiments, I am not able to understand in detail what is explained. There is one paragraph on the implication of their experimental result on the study of HS in the interstellar medium. This paragraph is more a discussion than a demonstration that their results will explain the observations as they do not go beyond the estimation of the branching ratio. These new data have not been included in a chemical model and so its impact is speculative.

Overall, such experimental results are crucial for astrochemical studies. These specific results can indeed be important and have an impact. The paper is however difficult to read for a general astrochemist not coming from this field of research. The very promising title of the paper is not completely reflecting the content of the paper.

Author Reply: This review invites us to revisit the paragraph relating to possible implications of the experimental measurements to studies of SH radicals in the interstellar medium and on the title of the paper. It is convenient to address these points along with those of reviewer 3 (below).

Reviewer #3

The article suggests to revise the relevant astrochemical models to justify the depletion of SH(X) in the interstellar medium. Based on the photolysis of jet cooled H₂S molecules in the 122 - 155 nm range the authors show that the quantum yield to form SH(X) decreases from 1 (at the longest excitation wavelengths) to zero at short wavelengths. Convoluting the interstellar radiation field taken from the literature with the total absorption cross-sections of H₂S the authors report that only 26% of all photoexcitation events should result in SH(X) fragments. The 26% is a very challenging value. What are the error estimates for this value? There are many theoretical spectroscopic studies that describe the VUV photo-dissociation dynamics of H₂S. This paper shows another UV photolysis of H₂S study but the obtained results cannot be used to estimate the SH radical density in the ISM as claimed by the author. Before publication the authors should pay attention on a few typos such as those on page 32 line 66 and in figure 4. I suggest also that they should change the title of this article and review the part concerning the SH density in the ISM. They should also link their study to the Chakraborty's work which was not mentioned and related to the sulfur isotopic fractionation in VUV photodissociation of H₂S at 121.6, 139.1, and 157.6 nm, the same wavelengths used in the present study (PNAS 2013 110 (44) 17650-17655).

Author Reply: The ~26% estimate is based on a convolution of several functions, each with poorly defined uncertainty, hence our use of the ~ symbol. The quoted uncertainty on σ is 10% (ref. 24), on the ISRF is 'within about 50%' (ref. 48) and on σ_{ion} is 10% (from ref. 15). The uncertainty on Γ is 12.5% - defined as the normalised (by range) root-mean-square error from the fitted sigmoidal

function. The root of the sum of the squares is thus 54%, which might suggest that only $26 \pm 14\%$ of all photoexcitation events should result in SH(X) fragments. But this is not a sensible approach. The uncertainty is reduced by recognizing that the uncertainty on σ_{ion} is only an issue in the narrow wavelength range above the first IP; at all longer wavelengths this uncertainty is zero. Further, the important uncertainties in σ_{ion} and in the standard model of the ISRF are not their absolute values, but their λ -dependences. These points are all now made in the revised paragraph (p. 10) reporting the quantum yield for forming SH(X) products when H₂S is photolyzed by the λ -dependent ISRF shown in Figure 4. Additionally, we have prepared and placed some additional files in the Supplementary Material that allow readers to convolute the existing H₂S absorption data and the present branching data with any user-chosen, λ -dependent ISRF.

We still consider that the original title provided a good description of the content of the paper and would be likely to attract the interest of the intended audience. However, since this title was questioned by two of the reviewers, we have considered this further and revised it to ‘Ultraviolet Photolysis of H₂S and its Implications for SH Radical Production in the Interstellar Medium’ which we hope is seen as providing a better description of the content.

The study by Chakraborty *et al.*, reports isotopic fractionation in solid sulfur films deposited following broadband (~ 2 nm), non-collision-free photolysis of slowly flowing samples of H₂S gas at wavelengths close to three of those used in the present study. The authors recognised the λ -dependent fractionation into SH(X) and SH(A) fragments and went on to assert that the SH fragments subsequently predissociate to yield elemental sulfur in all cases (p. 17653). In fact, the SH(X) fragments cannot predissociate (there is no lower lying dissociation limit), but it seems likely that they could undergo a further photoexcitation or react with another H₂S molecule during their time in the 60 cm long flow tube. So, the cause of the observed isotopic fractionation remains unclear – and not a topic appropriate for discussion in the present work – but the fact that prior photolysis experiments have been reported at similar VUV excitation wavelengths should have been noted and is now done so on p. 4 of the revised manuscript.

The reviewer identifies two typos. We have checked the typos in the paper.

We very much hope that you judge these responses to be appropriate and hope to see the manuscript published in a future issue of *Nature Communications*.

Yours sincerely,

K.J. Yuan

M.N.R. Ashfold

REVIEWERS' COMMENTS:

Reviewer #1 (Remarks to the Author):

The authors' response to my query regarding the branching ratio for dissociation in the region below the first ionization limit satisfies this issue. They now state the rationale for assuming 100% dissociation, and it is convincing. I also find their responses to the other two referees' questions to be thorough and convincing. I continue to recommend publication.

Reviewer #3 (Remarks to the Author):

I have nothing to add, the authors have answered most of the points underlined and the manuscript can be published as it is.

Reviewer #1 (Remarks to the Author):

The authors' response to my query regarding the branching ratio for dissociation in the region below the first ionization limit satisfies this issue. They now state the rationale for assuming 100% dissociation, and it is convincing. I also find their responses to the other two referees' questions to be thorough and convincing. I continue to recommend publication.

Reviewer #3 (Remarks to the Author):

I have nothing to add, the authors have answered most of the points underlined and the manuscript can be published as it is.

Author reply: Thank you very much for your comments. We are very glad to hear that the reviewers 1 and 3 find our manuscript publishable.